# Evaluation of a package of risk-based pharmaceutical and lifestyle interventions in patients with hypertension and/or diabetes in rural China: A pragmatic cluster randomised controlled trial

Xiaolin Wei[1], Zhitong Zhang[1], Marc K. C. Chong[2], Joseph P. Hicks[3], Weiwei Gong[4], Guanyang Zou[5], Jieming Zhong[4], John D. Walley[3], Ross E. G. Upshur[1], Min Yu[4]*

1 Dalla Lana School of Public Health, University of Toronto, Toronto, Ontario, Canada, 2 School of Public Health and Primary Care, Chinese University of Hong Kong, Hong Kong, China, 3 Nuffield Centre for International Health and Development, University of Leeds, Leeds, United Kingdom, 4 Zhejiang Provincial Centre for Disease Control and Prevention, Hangzhou, China, 5 Guangzhou University of Chinese Medicine, Guangzhou, China

* myu@cdc.zj.cn

## Abstract

### Background

Primary prevention of cardiovascular disease (CVD) requires adequate control of hypertension and diabetes. We designed and implemented pharmaceutical and healthy lifestyle interventions for patients with diabetes and/or hypertension in rural primary care, and assessed their effectiveness at reducing severe CVD events.

### Methods and findings

We used a pragmatic, parallel group, 2-arm, controlled, superiority, cluster trial design. We randomised 67 township hospitals in Zhejiang Province, China, to intervention (34) or control (33). A total of 31,326 participants were recruited, with 15,380 in the intervention arm and 15,946 in the control arm. Participants had no known CVD and were either patients with hypertension and a 10-year CVD risk of 20% or higher, or patients with type 2 diabetes regardless of their CVD risk. The intervention included prescription of a standardised package of medicines, individual advice on lifestyle change, and adherence support. Control was usual hypertension and diabetes care. In both arms, as usual in China, most outpatient drug costs were out of pocket. The primary outcome was severe CVD events, including coronary heart disease and stroke, during 36 months of follow-up, as recorded by the CVD surveillance system. The study was implemented between December 2013 and May 2017. A total of 13,385 (87%) and 14,745 (92%) participated in the intervention and control arms, respectively. Their mean age was 64 years, 51% were women, and 90% were farmers. Of all participants, 64% were diagnosed with hypertension with or without diabetes, and 36% were diagnosed with diabetes only. All township hospitals and participants completed the 36-

**Data Availability Statement:** According to the ethical agreements, the data is owned by the Zhejiang Provincial CDC, and available for research purpose of the study team. The data cannot be posted and downloaded in a public data depository, and not be able to be transmitted out of China due to the National Privacy Regulation. Only aggregated data can be shared upon request to: Department of Chronic Non-communicable Disease Prevention and Control at Zhejiang Provincial CDC, China (Email: zhejiangcdc_ncd@163.com).

**Funding:** The study was funded by Department for International Development, United Kingdom (https://www.gov.uk/government/organisations/department-for-international-development) (funding number: COMDIS-HSD, received by XW and JW). The study also received operating support from the Zhejiang Health Commission, China (https://wsjkw.zj.gov.cn/#, received by MY). XW and RU are also endowed Dalla Lana Chairs supported by the Dalla Lana School of Public Health at the University of Toronto (https://www.dlsph.utoronto.ca). The funders of the study had no role in study design, data collection, data analysis, data interpretation, or writing of the report.

**Competing interests:** The authors have declared that no competing interests exist.

**Abbreviations:** AOR, adjusted odds ratio; CHD, coronary heart disease; CVD, cardiovascular disease; GLMM, generalised linear mixed model; HICs, high-income countries; IRR, incidence rate ratio; ITT, intention-to-treat; LMICs, low- and middle-income countries; OR, odds ratio; SDG, Sustainable Development Goal; Zhejiang CDC, Zhejiang Provincial Centre for Disease Control and Prevention.

month follow-up. At 36 months, there were 762 and 874 severe CVD events in the intervention and control arms, respectively, yielding a non-significant effect on CVD incidence rate (1.92 and 2.01 per 100 person-years, respectively; crude incidence rate ratio = 0.90 [95% CI: 0.74, 1.08; $P = 0.259$]). We observed significant, but small, differences in the change from baseline to follow-up for systolic blood pressure (−1.44 mm Hg [95% CI: −2.26, −0.62; $P < 0.001$]) and diastolic blood pressure (−1.29 mm Hg [95% CI: −1.77, −0.80; $P < 0.001$]) in the intervention arm compared to the control arm. Self-reported adherence to recommended medicines was significantly higher in the intervention arm compared with the control arm at 36 months. No safety concerns were identified. Main study limitations include all participants being informed about their high CVD risk at baseline, non-blinding of participants, and the relatively short follow-up period available for judging potential changes in rates of CVD events.

## Conclusions

The comprehensive package of pharmaceutical and healthy lifestyle interventions did not reduce severe CVD events over 36 months. Improving health system factors such as universal coverage for the cost of essential medicines is required for successful risk-based CVD prevention programmes.

## Trial registration

ISRCTN registry ISRCTN58988083.

## Author summary

### Why was this study done?

- Pharmaceutical interventions have been shown to reduce cardiovascular disease (CVD) events, but the evidence has limited policy implications as medicines in trials have been typically been provided free of charge.

- Four previous trials examining CVD-risk-based management treatments in high-income countries were focused on healthy lifestyle interventions and did not identify any clear effects on CVD events. Two previous trials in low- and middle-income countries implemented both pharmaceutical and lifestyle interventions, but they were not designed to examine impacts on CVD events.

### What did the researchers do and find?

- We conducted a cluster randomised controlled trial in rural China among 31,326 patients with (1) hypertension who had a CVD risk of 20% or higher or (2) diabetes, and followed participants up for 36 months. Interventions included prescription of a standardised package of medicines, individual advice on lifestyle change, and adherence support provided by a team led by family doctors. We compared this to usual existing care.

- The study found that our comprehensive pharmaceutical and lifestyle interventions did not reduce severe CVD events.

- We observed significant, but small, differences in the change from baseline to follow-up of systolic blood pressure (−1.44 mm Hg) and diastolic blood pressure (−1.29 mm Hg) in the intervention arm compared to the control arm. Self-reported adherence to recommended medicines was significantly higher in the intervention arm compared with the control arm at 36 months.

- In the process evaluation, we identified that the unaffordability of the medications, caused by lack of health insurance coverage and the non-availability of cheaper generic medicine, appeared to discourage patients from taking recommended medicines.

### What do these findings mean?

- Comprehensive pharmaceutical and lifestyle interventions were feasible and effective in improving medicine uptake and reducing blood pressure, but not in achieving a reduction in severe CVD events, when implemented in a setting where the costs of essential medicines were insufficiently covered.

- This study highlights the importance of improving universal health coverage as a prerequisite for effective CVD prevention programmes in primary care in similar settings.

## Introduction

Cardiovascular disease (CVD) is the world's leading cause of mortality, representing 31% of all global deaths in 2016 [1]. The United Nations Sustainable Development Goal (SDG) 3 calls for a one-third reduction in CVD deaths by 2030. However, this goal will not be achieved without progressive prevention activities that include adequate control of blood pressure, glucose, and lipid levels, as well as smoking cessation, and reducing salt, alcohol, sugar, and trans fat intake [2]. Primary care provides the best setting for these solutions because hypertension and diabetes, the 2 most common risk factors for CVD, are managed by family doctors, who can provide continuous, coordinated, and comprehensive care. However, hypertension and diabetes are often treated and managed separately, as reflected in the frequent existence of separate guidelines and programmes. In addition, the 2 diseases are both typically inadequately controlled in low- and middle-income countries (LMICs) compared to high-income countries (HICs). Recent multi-country studies reported that only 10% of patients with hypertension had controlled hypertension in LMICs, compared to 50% in HICs [3,4].

To achieve SDG 3, a comprehensive treatment approach consisting of both pharmaceutical and lifestyle strategies is urgently needed in LMICs. The ability of recommended antihypertensive medicines, aspirin, and statins to prevent CVD events has been well documented in trials [1]. Recommended pharmaceutical therapy, often in the form of a combined polypill, can reduce CVD risk by 50%–60% in the long term [5], but the effects are often more modest, as shown in the PolyIran trial [6] (34% reduction in hazard ratio for major CVD events [95% CI: 20%, 45%] over 60 months). Meta-analyses of healthy lifestyle interventions, including those targeting weight loss, alcohol reduction, and smoking cessation, have shown moderate

improvements in biomarkers of CVD risk factors, such as blood pressure, body mass index (BMI), and total serum cholesterol [7]. Thus, combined pharmaceutical and lifestyle interventions could be more beneficial to patients at high risk of CVD. Several trials have reported positive effects of combined pharmaceutical and lifestyle interventions, including reduced patient CVD risks, blood pressure, and lipid profiles, but none demonstrated reductions in CVD events [8–11]. The strategy of risk-based pharmaceutical and lifestyle interventions has been suggested in national clinical guidelines; however, the policy question of whether this strategy has any impact to achieve SDG 3 remains unanswered due to insufficient evidence [12].

Our previous study identified health system barriers to sustainable CVD risk management, such as insufficient knowledge amongst family doctors, use of ineffective medicines, and lack of treatment support [13]. China's recent health reforms provided an opportunity for better control of non-communicable diseases as they strengthened primary care capacities and improved hypertension and diabetes management [14]. Based on this, we designed and implemented a comprehensive package of pharmaceutical and healthy lifestyle interventions embedded into routine primary care practice for primary prevention of CVD among patients with diabetes and hypertension. We have previously reported feasibility results from our pilot study [15], the trial protocol [16], and the effect on pharmaceutical management [17]. Here we report the evaluation of relevant interventions against the trial's primary and secondary outcomes during 36 months of follow-up.

## Methods

### Study design

Our study is a pragmatic, parallel group, 2-arm, cluster randomised, controlled, superiority trial. It aims to assess the effectiveness of a package of comprehensive pharmaceutical and healthy lifestyle interventions targeting primary prevention of CVD among patients with high CVD risk in rural China. We conducted the research in township hospitals, which are rural primary care facilities. Township hospitals and their catchment populations are the clusters of the trial. Details of the trial design and analysis plan have been published elsewhere [16]. In brief, we selected all township hospitals, except 1 that served as the pilot site, in 3 counties in Zhejiang Province. Cluster eligibility criteria included having electronic medical records covering the last 2 years for residents in these townships. This study was approved by the Ethics Review Board of the University of Leeds, UK (reference HSLTLM/12/010) and the Ethics Committee of Zhejiang Provincial Centre for Disease Control and Prevention, China (reference 18/06/2012). Written informed consent was obtained from all participating individuals. We used the CONSORT cluster trials checklist (S1 CONSORT Checklist) to help in reporting the study [18].

### Participants

We recruited participants from township hospitals based on collected health records from all residents in their catchment areas. Patients were considered eligible if they had the following characteristics: (1) aged 50 to 74 years; (2) permanent residents in the township; (3) having either hypertension with a 10-year CVD risk $\geq$ 20% calculated using the Asian equation [19], or type 2 diabetes (with or without hypertension); (4) free from any diagnosed mental diseases or physical disabilities, any history of severe CVD events, or other diseases as defined in the protocol [16]; (5) not currently hospitalised or living in a long-term care facility; (6) no serious adverse effects to the recommended medicines, and (7) having a diastolic blood pressure $\geq$ 60 mm Hg. After cluster randomisation, all potentially eligible participants were invited to visit the township hospital, where family doctors explained the study, recruited patients, and obtained their consent.

## Randomisation and masking

In December 2013, we randomised a total of 67 township hospitals, without stratification, to the intervention and control arms in a 34:33 ratio, via an independent statistician using computer-generated random numbers. All eligible consenting participants received either the intervention or control treatment based on the treatment allocated to their township hospital. Due to the nature of the interventions, health providers and patients could not be blinded, but the analysis of the trial was blinded.

## Procedures

In both arms patients had recurring booked consultations with their doctors at least quarterly. All recommended medicines were available in township hospitals or pharmacies in both arms. Patients purchased their medications from township hospitals or pharmacies using prescriptions from their family doctors, and then reported their adherence to the prescriptions in the next consultation.

In the control arm, usual care of hypertension and diabetes continued per existing practice. No additional recommendations for pharmaceutical or healthy lifestyle modifications were provided. The 2010 Chinese national hypertension [20] and diabetes [21] guidelines were available, but they were often not referred to in primary care consultations because the guidelines contain large amounts of information for tertiary care [22]. Family doctors treated hypertension and diabetes according to their own discretion and group practice. No specific training regarding CVD risk management was provided. Healthy lifestyle change education was given based on controlling either hypertension or diabetes, but compared to the intervention arm this was not using a holistic approach to CVD risk reduction. Treatment adherence support was not provided.

In the intervention arm, patients received prescriptions and healthy lifestyle education for treatment of their existing medical conditions and CVD prevention. All patients with hypertension, or hypertension and diabetes, were prescribed a standard combination [23] of (1) 2 antihypertensives (2 different kinds selected from thiazide diuretics, calcium channel blockers, angiotensin-converting enzyme inhibitors, angiotensin II receptor blockers, and beta-blockers) [24], (2) a statin, and (3) a low dose aspirin, unless contraindicated. Patients with only diabetes received the same pharmaceutical package but with only 1 antihypertensive, plus their anti-diabetic medicines, if any. Patients already on other older antihypertensive medicines were advised to switch to the recommended medicines in standardised packages. Patients who had a history of, or showed signs of, gastrointestinal bleeding were not prescribed aspirin. During consultation, family doctors provided individualised health education focusing on smoking cessation, salt reduction, and reduction of alcohol consumption depending on the patient's situation. A family treatment supporter was selected at the patient's home to support him/her taking medicines and adhering to lifestyle changes. At the facility level, we provided annual training to all family doctors using our CVD risk management guidelines, which covered (1) prescribing for primary prevention of CVD, (2) personalised advice for healthy lifestyle changes, and (3) advice to improve adherence [16]. Township hospitals held monthly meetings to discuss doctors' experience of using the guidelines and chronic disease management (Table 1).

## Data collection

On enrolment, family doctors measured participants' blood pressure using a standardised mercury sphygmomanometer after 5 minutes of seated rest. They also measured their body weight and height, and asked for details on their medication history. These measures were

**Table 1. Intervention strategies to reduce the risk of cardiovascular disease (CVD).**

| Target | Intervention strategies in the intervention arm | Usual care in the control arm |
|---|---|---|
| Family doctors | **Operational guidelines:** The guidelines contained a definition of high risk of CVD, workflow chart for patient management, how to diagnose hypertension and type 2 diabetes, suggested prescriptions and CVD risk management practices, lifestyle evaluation and recommendations, treatment support, and managing the side effects of medicines. In addition, we also incorporated elements of China's national hypertension and type 2 diabetes guidelines for township hospitals. **Training workshops:** Training workshops were provided on an annual basis by local senior doctors based on the operational guidelines, including hypertension/diabetes management, prevention, and communication skills, using lectures, health education videos, case discussions, role plays, and question and answer sessions. **Monthly meetings in township hospitals for performance monitoring:** A senior doctor led the review of the implementation of intervention strategies, and provided comments to related questions on CVD risk management and treatment of patients by doctors. | **Guidelines and training:** National hypertension and type 2 diabetes guidelines were available but were not referred to in practice because the guidelines focus on tertiary care. No specific training/guidelines were given on reducing CVD risks based on a holistic approach. **Training workshops:** Routine annual training workshops on hypertension and type 2 diabetes management were provided, but there was no research-driven systematic training provided on CVD risk management. **Monthly meetings:** Routine monthly internal meetings were held, but there were no specific discussions dedicated to CVD risk management. |
| Patients and their treatment supporters | **Recommended medicines:** A combination of 2 antihypertensives, a statin, and a low dose of aspirin were recommended to patients with hypertension. One antihypertensive, a statin, and a low dose of aspirin were recommended for patients with type 2 diabetes but not hypertension. Anti-diabetic medicine or insulin was given for diabetic patients if necessary. **Health education messages from family doctors during regular follow-up consultations (from monthly to quarterly):** In their consultations, family doctors gave targeted health education messages to participants based on their health conditions, e.g., smoking cessation messages to smokers, and messages regarding the benefits of the combined medicines to those who did not adhere to prescriptions. **Enhanced follow-up appointment:** Participants attended follow-up consultations in township hospitals at least quarterly with their family doctors. Nurses communicated with participants monthly through phone calls/SMS messages. **Treatment supporter:** Family doctors or nurses helped to identify and train a treatment supporter (normally a family member) who reminded patients to take medicines, maintain a healthy lifestyle, and adhere to their follow-up consultations. | **Recommend medications and health education:** Treatment and lifestyle changes were offered according to existing knowledge and at the individual clinician's discretion, but based on either hypertension or type 2 diabetes instead of a holistic approach for CVD prevention. In practice, most doctors prescribed only 1 antihypertensive for hypertensive patients, and no antihypertensive for diabetic patients. Very few patients were given a statin or aspirin. Anti-diabetic medicine or insulin was given for diabetic patients if necessary. Existing general health education messages were provided. **Follow-up appointment:** Patients with hypertension or type 2 diabetes were followed up at least quarterly by family doctors according to national guidelines. **Treatment supporter:** The option of a treatment supporter was not provided. |
| Medicines | Modern medicines were available in both the intervention and control arms, and their prices were the same in the 2 arms. | |

repeated on a quarterly basis when participants were followed up in township hospitals. All information was recorded in an internet-based public health management information system. The trial stopped 36 months (May 2017) after enrolment of the last participant based on our protocol.

All acute severe CVD events were collected through Zhejiang CDC's surveillance system [25] based on the adapted WHO MONICA definitions [26] compatible with ICD-10 codes. These included coronary heart disease (CHD) and stroke, where CHD includes acute myocardial infarction, ischaemic cardiac arrest, and unclassifiable CHD deaths, and stroke includes haemorrhagic stroke (e.g., subarachnoid and intracerebral), ischaemic stroke/infarction (e.g., thrombosis and embolism), and unclassifiable stroke [27]. We did not include minor CVD events such as angina and transient ischaemic attacks because they are often unrecorded. To be eligible as endpoints, all CVD events had to be verified from hospital records, and were reviewed and verified by centres for disease control at the county and provincial levels. All reported CVD events were then also been verified by primary care facilities through home visits. The process was supervised under the Validation Committee in the Zhejiang CDC. We previously reported on the data collection, reporting, and validation of the Zhejiang CDC's surveillance system [25]. In addition, family doctors asked participants if they had experienced

any stroke or heart attack events that were diagnosed by hospitals at the 12th, 24th, and 36th month following randomisation; if any, the CVD events were reported to the Validation Committee and then verified and included in the records. Participants were classified as lost to follow-up if they could not be contacted after 3 attempts (via telephone, message, or home visit) and their CVD status was unknown by the 36th month.

## Outcomes

All outcomes were measured at the patient level. The primary outcome was the number of severe CVD events recorded by Zhejiang's CVD surveillance system [25]. We also collected the following secondary outcomes: (1) mortality due to severe CVD event, (2) number of CHD events, (3) mortality due to CHD event, (4) number of stroke events, (5) mortality due to stroke, (6) time to the first reported severe CVD event, (7) time to mortality due to CVD, (8) change in diastolic and systolic blood pressure (mm Hg) between baseline and 36 months, (9) adherence to the final quarterly follow-up consultation at 36 months, and (10) self-reported adherence at 36 months to 2 antihypertensive drugs for patients with hypertension or 1 antihypertensive drug for patients with only diabetes, aspirin, and statin. We also recorded any minor or serious adverse events. We added secondary outcomes 2–5 to identify any changes in CVD subgroups. In subsequent papers, we will report our other protocol-defined secondary outcomes from our panel data and cost-effectiveness analysis, which are based on different datasets as per our protocol [16]. We also previously reported feasibility measures in our process evaluation paper [28].

## Statistical analysis

We calculated the necessary sample size as 32 clusters per arm and 450 patients per cluster, to have 90% power to detect a 20% reduction in the severe CVD event incidence rate after 36 months, based on an assumed CVD event rate of 5% in the control arm and a coefficient of variation of 0.15, with hypothesis testing using a 2-sided $P$ value with a 5% threshold for statistical significance.

We calculated the exact follow-up time for each participant until the end of the trial or death. We estimated crude and covariate-adjusted ("adjusted") estimates of the treatment effect on all outcomes using generalised linear mixed models (GLMMs). To estimate the crude treatment effect (as an incidence rate ratio [IRR]) on our primary outcome, we used a GLMM with a fixed effect for treatment arm, a random intercept for cluster, and an "offset" variable for the log of person follow-up time, which used a log-link with Poisson errors. To estimate the adjusted treatment effect, we repeated the same model but included patient-level variables as per table footnotes. We also estimated crude and adjusted treatment effects for all our secondary outcomes using a variety of mixed-effects models, with the same random and fixed effects (unless stated: see relevant table footnotes) included. We employed GLMMs with Poisson errors and log-links (if no evidence of overdispersion) or GLMMs with negative binomial distributions and log-links (with treatment effects estimated as IRRs) to analyse event and mortality outcomes, Cox proportional hazard mixed-effects models (with treatment effects estimated as HRs) for time to event outcomes, GLMMs with normal errors and identity links (with treatment effects estimated as mean differences) for blood pressure outcomes, and GLMMs with binomial errors and logistic links (with treatment effects estimated as odds ratios [ORs]) for adherence and adverse event outcomes. The intra-cluster correlation coefficient for the primary outcome was calculated by dividing the between-cluster variance by the total variance. The variance terms were estimated by fitting the relevant (Poisson/negative binomial) unconditional mixed-effects model with a random intercept for cluster [29,30]. All 95%

confidence intervals and *P* values for GLMMs were based on either the Wald statistic (non-normal errors) or the *t* statistic (normal errors). We analysed all data using SAS statistical software (PROC MIXED and PROC GLIMMIX functions). Where any patients were missing outcomes or covariates in adjusted analyses, we excluded these patients and performed complete case analyses. We also ran cluster-level analyses for all our outcomes, as reported in S1 Text.

Our primary analysis was of the intention-to-treat (ITT) population and included all participants within all clusters as originally recruited and randomised, including all participants who left the trial after their first consultation or moved out of their cluster catchment area, and those who were wrongly classified as eligible: They all remained in the province so their CVD status was still tracked by the surveillance system. We also defined a "modified ITT" analysis population, which included all clusters and patients as originally recruited and randomised but excluding any patients with missing outcome or covariate data. We also repeated our analyses for all outcomes reported here in a per-protocol analysis population using the same methods described above. This per-protocol analysis population included all clusters as originally recruited and randomised, and across both arms it included all eligible patients except those patients who left the trial within the first quarter or those who were wrongly recruited. In addition, in the intervention arm this population excluded patients who did not have at least 50% adherence to their prescriptions across all their follow-up consultations. This included taking recommended antihypertensives (specifically, patients with hypertension taking a standard combination of 2 antihypertensives, and patients with diabetes only taking 1 antihypertensive) and statins or aspirin. These per-protocol analyses were designed after the trial because we observed a low rate of adherence to prescriptions, beyond what we expected.

We did a range of exploratory subgroup analyses of the primary outcome based on the pre-randomisation baseline characteristics of diagnosis (hypertension or hypertension and diabetes versus diabetes only), patient income level (<10,000 RMB [<US$1,644] versus ≥10,000 RMB [≥US$1,644] per annum), and education (primary school or below versus high school or above). In our analyses of secondary outcomes and in our per-protocol and subgroup analyses, we did not adjust the *P* values or confidence intervals for multiple testing, and we treat the results as exploratory.

## Results

We randomised 34 township hospitals to the intervention arm and 33 township hospitals to the control arm. Across all hospitals' catchment areas, based on electronic health records, we identified a total of 31,326 potentially eligible participants: 15,380 in the intervention arm and 15,946 in the control arm. At the recruitment stage (December 2013 to May 2014), 1,995 and 1,201 eligible participants declined to participate in the intervention and control arms, respectively, resulting in recruitment of 13,385 (87%) and 14,745 (92%) participants in the intervention and control arms, respectively (Fig 1).

As shown in Table 2, participants in the 2 arms were similar with respect to baseline characteristics. The mean age was 64 years, 51% were women, 90% were farmers, and 77% had a primary education or below. The median annual per capita income was US$1,424, reflecting the poor and rural setting. Of all participants, 64% were diagnosed with hypertension with or without diabetes, and 36% were diagnosed with diabetes only. Following their first consultation, 1,144 and 1,694 participants in the intervention and control arms, respectively, chose to stop participating, and 76 and 83 participants in the intervention and control arms, respectively, were found to have pre-existing CVD (and therefore ineligible), but all were included in the ITT analysis population. During the follow-up there were 366 and 483 all-cause mortalities, and 135 and 194 participants moved out of their cluster catchment areas, in the intervention and control

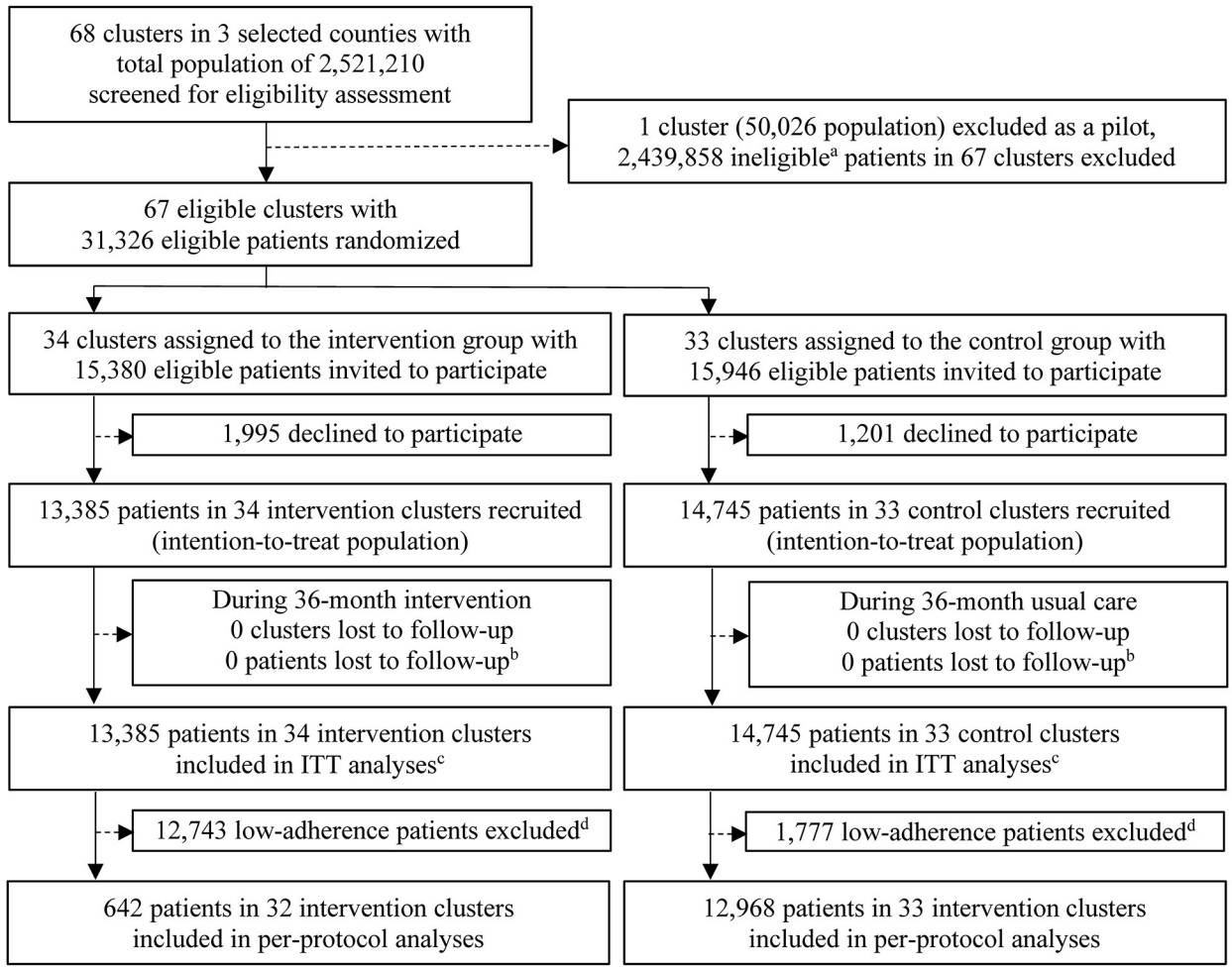

**Fig 1. Trial profile.** [a]Ineligible patients included 2,419,058 who didn't meet the eligibility criteria (aged 50–74 years, with a calculated 10-year CVD risk $\geq$ 20% and having hypertension, or diagnosed with type 2 diabetes); 222 who had mental diseases; 492 who had cancers; 2,076 who had pre-existing acute coronary heart disease or had experienced a stroke according to heath records; 293 whose diastolic blood pressure was <60 mm Hg; 16,562 who were out of town for more than 3 months or were non-locals/urban residents; and 1,155 who couldn't take the recommended medicines due to other severe diseases. [b]Patients whose records of acute cardiovascular diseases or death could not be obtained at the end of the 36-month follow-up period were categorised as lost to follow-up. [c]Patients who died (intervention = 366, control = 483), moved out of cluster catchment areas (intervention = 135, control = 194), were subsequently found to have pre-existing acute cardiovascular diseases not identified through initial health records (intervention = 76, control = 83), or subsequently decided to leave the trial within the first quarter of the follow-up period (intervention = 1,144, control = 1,694) are all included in the main (covariate unadjusted) intention-to-treat (ITT) analyses. [d]Patients excluded from the per-protocol analysis: (1) patients in the intervention group who did not report taking both suggested hypertensive medicines and a statin/aspirin for more than half of the follow-up period before the occurrence of acute cardiovascular disease, if they had an occurrence, or until the end of follow-up, if they did not ($n$ = 11,523); (2) patients who were subsequently found to have pre-existing acute cardiovascular diseases not identified through initial health records (intervention = 76, control = 83); and (3) patients who subsequently decide to leave the trial within the first quarter of the follow-up period (intervention = 1,144, control = 1,694).

arms, respectively (Fig 1). All clusters and participants were successfully followed up based on electronic medical records and/or CVD events reported via family doctors, allowing us to have complete data for the primary and secondary outcomes for the ITT population. However, there were some missing data for the adherence and blood pressure outcomes, and for the covariates in adjusted analyses. Missing case details are provided in the footnotes of the results tables.

Our interventions did not have a statistically significant effect on the primary outcome. Among all participants, 762 and 874 severe CVD events were reported in the intervention and

**Table 2. Characteristics of the 28,130 participants in the trial.**

| Patient characteristics | Intervention group (n = 13,385) | Control group (n = 14,745) | Total (n = 28,130) |
|---|---|---|---|
| Number of clusters | 34 | 33 | 67 |
| Age (years), mean (SD) | 64.3 (6.3) | 64.2 (6.1) | 64.3 (6.1) |
| Sex, n (%) | | | |
| Male | 6,443 (48.1%) | 7,316 (49.6%) | 13,759 (48.9%) |
| Female | 6,942 (51.9%) | 7,429 (50.4%) | 14,371 (51.1%) |
| Body mass index (kg/m$^2$), mean (SD) | 24.0 (3.0) | 23.9 (2.9) | 23.9 (2.9) |
| Occupation, n (%) | | | |
| Farmer | 11,735 (89.9%) | 12,960 (90.2%) | 24,695 (90.0%) |
| Worker | 205 (1.6%) | 181 (1.3%) | 386 (1.4%) |
| Technician | 37 (0.3%) | 55 (0.4%) | 92 (0.3%) |
| Administrative staff | 18 (0.1%) | 8 (0.1%) | 26 (0.1%) |
| Service worker | 58 (0.4%) | 56 (0.4%) | 114 (0.4%) |
| Personal business | 197 (1.5%) | 165 (1.2%) | 362 (1.3%) |
| Retired | 105 (0.8%) | 169 (1.2%) | 274 (1.0%) |
| Other | 702 (5.4%) | 778 (5.4%) | 1,480 (5.4%) |
| Educational level, n (%) | | | |
| Primary school or below | 10,356 (77.7%) | 11,070 (75.4%) | 21,426 (76.5%) |
| High school | 2,691 (20.2%) | 3,234 (22.0%) | 5,925 (21.1%) |
| College or above | 290 (2.1%) | 382 (2.6%) | 672 (2.4%) |
| Marital status, n (%) | | | |
| Married | 11,813 (95.4%) | 13,109 (95.6%) | 24,922 (95.5%) |
| Single, divorced, or widowed | 566 (4.6%) | 610 (4.4%) | 1,176 (4.5%) |
| Annual per capita income (US dollars), median (IQR) | 1,424 (712, 1,994)[a] | 1,424 (712, 2,374)[a] | 1,424 (712, 2,136)[a] |
| Diagnosis, n (%) | | | |
| Hypertension with/without type 2 diabetes | 8,800 (65.7%) | 9,224 (62.6%) | 18,024 (64.1%) |
| Type 2 diabetes without hypertension | 4,585 (34.2%) | 5,521 (37.4%) | 10,106 (35.9%) |
| 10-year CVD risk in patients with hypertension, median (IQR) | 26.5% (23.0%, 30.8%) | 26.3% (22.9%, 30.8%) | 26.4% (22.9%, 30.8%) |

[a]The recommended package of medicines cost US$35–US$44 per month in the local setting, i.e., US$420–US$528 per year.

control arms, respectively. The majority (99.4%) of CVD events were collected through Zhejiang's CVD surveillance system, while 4 and 5 CVD events from the intervention and control arms, respectively, were collected via family doctors at their consultations. The observed incidence rate for severe CVD events was 1.92 and 2.01 per 100 person-years in the intervention and control arms respectively, and the crude IRR was 0.90 (95% CI: 0.74, 1.08; P = 0.259; Table 3). The primary outcome's IRR was not substantially altered when adjusting for covariates (Table 3). There was no significant treatment effect on any of the secondary event, mortality, or time to event outcomes, including CVD mortality (Tables 3 and 4). The results from our adjusted analyses were consistent with the crude results for all secondary outcomes (Tables 3 and 4). The cluster-level analyses of both primary and secondary outcomes produced results consistent with those of the patient-level analyses (Tables A–C in S1 Text).

Our per-protocol population included just 642 participants in the intervention arm and excluded 2 clusters in the intervention arm that had no remaining participants, while retaining 12,968 patients and all clusters in the control arm. There were no significant differences between the 2 arms regarding severe CVD events (1.58 versus 1.94 per 100 person-years in the intervention and control groups, respectively; crude IRR = 0.81 [95% CI: 0.55, 1.20;

**Table 3. Comparison of primary and secondary outcomes for disease event rates between the intervention and control arms (individual-level data analyses).**

| Outcome | Intervention | | Control | | Crude IRR (95% CI)[b] | P value[b] | Adjusted IRR (95% CI)[c] | P value[c] |
|---|---|---|---|---|---|---|---|---|
| | N (%)[a] | Event rate | N (%)[a] | Event rate | | | | |
| CVD event[d] | 762 (5.69%) | 1.92 | 874 (5.93%) | 2.01 | 0.90 (0.74 to 1.08) | 0.259 | 0.87 (0.73 to 1.05) | 0.148 |
| CVD mortality | 165 (1.23%) | 0.42 | 185 (1.25%) | 0.42 | 0.91 (0.69 to 1.21) | 0.516 | 0.90 (0.68 to 1.18) | 0.455 |
| CHD event | 92 (0.69%) | 0.23 | 120 (0.81%) | 0.28 | 0.84 (0.63 to 1.11) | 0.217 | 0.86 (0.65 to 1.14) | 0.290 |
| CHD mortality | 43 (0.32%) | 0.11 | 55 (0.37%) | 0.13 | 0.80 (0.46 to 1.37) | 0.410 | 0.73 (0.39 to 1.37) | 0.331 |
| Stroke event | 659 (4.92%) | 1.66 | 738 (5.01%) | 1.69 | 0.91 (0.75 to 1.11) | 0.359 | 0.88 (0.73 to 1.07) | 0.200 |
| Stroke mortality | 122 (0.91%) | 0.31 | 132 (0.90%) | 0.30 | 0.99 (0.75 to 1.30) | 0.932 | 0.99 (0.77 to 1.28) | 0.937 |

CHD, coronary heart disease; CVD, cardiovascular disease; IRR, incidence rate ratio. Intervention, $n$ = 13,385; control, $n$ = 14,745. The total person-years of follow-up were 39,657 and 43,557 for the intervention group and control group, respectively.

[a]Individual-level summary statistics: $N$ is the number of events; percent is the total number of events/total number of individuals (per arm). Event rate is per 100 person-years. For all disease event outcomes (as opposed to disease mortality outcomes), individuals may experience >1 event, and so the $N$ (%) statistics are based on the occurrence/non-occurrence of the event per person only.

[b]The intervention minus control crude IRRs were obtained via a Poisson mixed-effects model (primary outcome) or a Poisson or negative binomial mixed-effects model (secondary outcomes: depending on the level of dispersion) with a single fixed effect of treatment arm and a random intercept for cluster. The 95% CIs and $P$ values were calculated based on the Wald statistic. In all crude analyses there were no missing patients.

[c]The intervention minus control adjusted IRRs were obtained via a Poisson mixed-effects model (primary outcome) or a Poisson or negative binomial mixed-effects model (secondary outcomes: depending on the level of dispersion) with fixed effects for treatment arm, patient age, sex, body mass index, occupation (farmer or non-farmer), educational level (primary school or below, or high school or above), marital status (married or non-married), income (less than 10,000 RMB or 10,000 RMB or above), and diagnosis (hypertension without type 2 diabetes, type 2 diabetes without hypertension, or hypertension with type 2 diabetes), and a random intercept for cluster. The 95% CIs and $P$ values were calculated based on the Wald statistic. In all adjusted analyses, 1,112 (8.3%) and 1,432 (9.7%) patients in the intervention and control arms, respectively, were excluded due to missing outcome or covariate data.

[d]Only 14 and 19 patients in the intervention and control arms, respectively, reported a second severe CVD event, and none reported more than 2 severe CVD events. The intra-cluster correlation coefficient for the primary CVD event outcome was 0.024, and was calculated by dividing the between-cluster variance by the total variance. The variance terms were estimated by fitting the relevant (Poisson or negative binomial) unconditional mixed-effects model with only a random intercept for cluster.

$P$ = 0.298]). When analysing the per-protocol population there were also no significant results in our secondary outcomes of mortality and events (Tables D–F in S1 Text).

**Table 4. Comparison of time to first CVD event and time to CVD mortality outcomes between the intervention and control arms (individual-level data analyses).**

| Outcome | Mean (SD) number of days[a] | | Crude HR (95% CI)[b] | P value[b] | Adjusted HR (95% CI)[c] | P value[c] |
|---|---|---|---|---|---|---|
| | Intervention | Control | | | | |
| Time to the first CVD event | 554.2 (311.4) | 554.8 (314.8) | 0.87 (0.72 to 1.05) | 0.140 | 0.86 (0.72 to 1.03) | 0.108 |
| Time to CVD mortality | 614.8 (314.8) | 627.4 (307.2) | 0.92 (0.69 to 1.22) | 0.543 | 0.91 (0.69 to 1.19) | 0.478 |

CVD, cardiovascular disease; HR, hazard ratio. Intervention, $n$ = 13,385; control, $n$ = 14,745.

[a]Individual-level summary data are presented as mean (SD) number of days until the outcome, excluding censored outcomes.

[b]The intervention minus control crude HRs were obtained using a mixed-effects Cox proportional hazard model with a single fixed effect of treatment arm and a random intercept for cluster. The 95% CIs and $P$ values were calculated based on the Wald statistic. In both crude analyses there were no missing patients.

[c]The intervention minus control adjusted HRs were obtained using a mixed-effects Cox proportional hazard model with fixed effects for treatment arm, patient age, sex, body mass index, occupation (farmer or non-farmer), educational level (primary school or below, or high school or above), marital status (married or non-married), income (less than 10,000 RMB or 10,000 RMB or above), and diagnosis (hypertension without type 2 diabetes, type 2 diabetes without hypertension, or hypertension with type 2 diabetes), and a random intercept for cluster. The 95% CIs and $P$ values were calculated based on the Wald statistic. In the adjusted analysis of the time to first CVD event outcome, 1,153 (8.6%) and 1,470 (10.0%) patients in the intervention and control arms, respectively, were excluded due to missing covariate data, and in the adjusted analysis of the time to CVD mortality outcome, 1,644 (12.3%) and 2,037 (13.8%) patients in the intervention and control arms, respectively, were excluded due to missing covariate data.

**Table 5. Comparison of systolic and diastolic blood pressure outcomes between intervention and control arms (individual-level data analyses).**

| Outcome | Mean (SD) blood pressure (mm Hg) | | Crude mean difference (95% CI)[a] | P value[a] | Adjusted mean difference (95% CI)[b] | P value[b] |
|---|---|---|---|---|---|---|
| | Intervention | Control | | | | |
| **Systolic blood pressure** | | | | | | |
| Baseline | 136.2 (11.7) | 135.9 (11.4) | | | | |
| Endline | 132.4 (9.34) | 133.4 (9.71) | −1.44 (−2.26 to −0.62) | <0.001 | −1.50 (−2.30 to −0.70) | <0.001 |
| **Diastolic blood pressure** | | | | | | |
| Baseline | 81.7 (6.81) | 81.8 (6.67) | | | | |
| Endline | 79.5 (6.31) | 80.6 (6.30) | −1.29 (−1.77 to −0.80) | <0.001 | −1.31 (−1.79 to −0.83) | <0.001 |

Intervention, $n$ = 13,385; control, $n$ = 14,745.

[a]The intervention minus control crude mean endline differences were obtained via a linear mixed-effects model with a single fixed effect of treatment arm and a random intercept for cluster. The 95% CI and $P$ values were calculated based on the $t$ statistic. In both crude analyses there were no missing patients.

[b]The intervention minus control adjusted mean endline differences were obtained via a linear mixed-effects model with fixed effects for treatment arm, patient age, sex, body mass index, occupation (farmer or non-farmer), educational level (primary school or below, or high school or above), marital status (married or non-married), income (less than 10,000 RMB or 10,000 RMB or above), and diagnosis (hypertension without type 2 diabetes, type 2 diabetes without hypertension, hypertension with type 2 diabetes), and a random intercept for cluster. The 95% CI and $P$ values were calculated based on the $t$ statistic. In the adjusted analysis of the systolic blood pressure outcome, 2,624 (19.6%) and 3,339 (22.6%) patients in the intervention and control arms, respectively, were excluded due to missing covariate or outcome data, and in the adjusted analysis of the diastolic blood pressure outcome 2,620 (19.6%) and 3,338 (22.6%) patients in the intervention and control arms, respectively, were excluded due to missing covariate or outcome data.

We did observe significant crude differences in the intervention arm compared to the control arm in the change from baseline to 36 months regarding systolic blood pressure (−1.44 mm Hg [95% CI: −2.26, −0.62; $P$ < 0.001]) and diastolic blood pressure (−1.29 mm Hg [95% CI: −1.77, −0.80; $P$ < 0.001]), with similar adjusted results (Table 5). In our per-protocol analyses we observed a similarly sized significant difference in diastolic blood pressure change (−1.25 mm Hg [95% CI: −2.02, −0.49; $P$ = 0.001]), but a non-significant difference in systolic blood pressure change (−0.64 mm Hg [95% CI: −1.88, −0.60; $P$ = 0.315]; Table G in S1 Text).

Fig 2 illustrates that adherence to recurring booked quarterly consultations remained high in both arms throughout the trial, only appearing to decline slightly from 4 months until 36 months in both arms. Correspondingly, there was no significant difference between arms in patients' adherence to their final quarterly consultations (adjusted OR [AOR] = 1.44 [95% CI: 0.88, 2.36; $P$ = 0.144]). In total, 84.9% (11,359) of patients in the intervention arm had a treatment supporter, compared with only 0.4% (63) of patients in the control arm (AOR = 159.32, [95% CI: 124.53, 203.81; $P$ < 0.001]). In most cases, treatment supporters were patients' spouses and children. Among patients with hypertension, either alone or with diabetes, as shown in Fig 3A and 3B, only 22.5% reported adherence to 2 antihypertensive drugs at baseline in both arms. This then steadily increased to 37.2% at 36 months in the intervention arm compared to 16.8% in the control arm (AOR = 4.75 [95% CI: 3.82, 5.91; $P$ < 0.001]). In the intervention arm, patients reported significant improvement in adherence to aspirin (1% at baseline and 16.7% at 36 months) and statins (0.4% at baseline and 8% at 36 months), compared to consistently low levels of adherence reported in the control arm (aspirin, 1% at baseline and 1.7% at 36 months; AOR = 12.1 [95% CI: 7.69, 18.9; $P$ < 0.001]; and statin, 0.4% at baseline and 0.8% at 36 months; AOR = 9.69 [95% CI: 5.65, 16.6; $P$ < 0.001]). Among patients with only diabetes (Fig 3C and 3D), we did not identify a significant difference between the 2 arms in adherence to 1 antihypertensive at 36 months (AOR = 1.31 [95% CI: 0.93, 1.84; $P$ = 0.13]). Their adherence to aspirin was around 1% at baseline in both arms, but this increased substantially in the intervention arm (to 8.9%) but not in the control arm (1.1%),

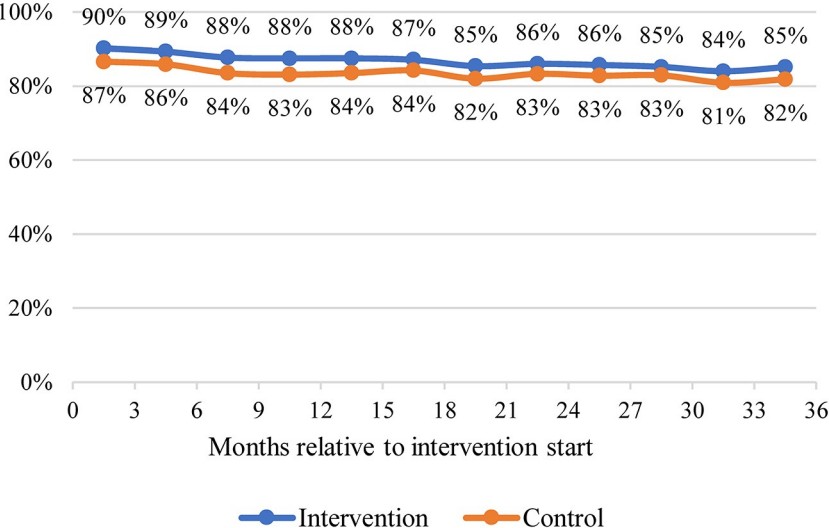

**Fig 2. Adherence to booked quarterly consultations.** Adherence to booked consultations is calculated as the proportion of patients in each arm who responded to their quarterly (i.e., every 3 months) follow-up clinical consultations with health workers within each 3-month follow-up period. A mixed-effects logistic regression analysis of patients' adherence (yes/no) to their final quarterly booked consultation—with fixed effects for treatment arm, patient age, sex, body mass index, occupation (farmer or non-farmer), educational level (primary school or below, or high school or above), marital status (married or non-married), income (less than 10,000 RMB or 10,000 RMB or above), and diagnosis (hypertension without type 2 diabetes, type 2 diabetes without hypertension, hypertension with type 2 diabetes), and a random intercept for cluster—demonstrated no significant difference between treatment arms (intervention versus control adjusted odds ratio = 1.44 [95% CI: 0.88, 2.36; $P$ = 0.144]). This analysis excluded 1,112 (8.3%) and 1,432 (9.7%) patients from the intention-to-treat population intervention ($n$ = 13,385) and control ($n$ = 14,745) arms, respectively, due to missing covariate or outcome data.

reflecting a significantly higher level of adherence in the intervention arm at 36 months (AOR = 9.09 [95% CI: 5.44, 15.2; $P$ < 0.001]). For these patients, there was a similar pattern observed for statin adherence, with near 0% adherence at baseline in both arms but 6.1% adherence in the intervention arm, compared to 0.4% in the control arm, at 36 months (AOR = 12.8 [95% CI: 6.68, 24.5; $P$ < 0.001]).

We analysed patients' self-reported adherence (yes/no) to at least 1 or 2 antihypertensive medicines, aspirin, and/or a statin at 36 months of follow-up via mixed-effects logistic regression models, with fixed effects for treatment arm, patient age, sex, body mass index, occupation (farmer or non-farmer), educational level (primary school or below, or high school or above), marital status (married or non-married), income (less than 10,000 RMB or 10,000 RMB or above), and diagnosis (hypertension without type 2 diabetes, type 2 diabetes without hypertension, hypertension with type 2 diabetes), and a random intercept for cluster. In the subgroup of patients with hypertension (intervention $n$ = 8,800, control $n$ = 9,224), self-reported adherence to at least 2 antihypertensive drugs, aspirin, and/or a statin at 36 months was significantly higher in intervention arm patients compared to control arm patients (AOR for self-reported adherence to at least 2 antihypertensive medicines at 36 months = 4.75 [95% CI: 3.82, 5.91; $P$ < 0.001]; AOR for self-reported adherence to aspirin at 36 months = 12.1 [95% CI: 7.69, 18.9; $P$ < 0.001]; AOR for self-reported adherence to a statin at 36 months = 9.69 [95% CI: 5.65, 16.6; $P$ < 0.001]). In all these subgroup analyses, 851 (9.7%) and 1,094 (11.9%) patients in the intervention and control arms, respectively, were excluded due to missing covariate or outcome data. In the subgroup of patients with type 2 diabetes only (intervention $n$ = 4,585, control $n$ = 5,521), self-reported adherence to at least 1 antihypertensive medicine at

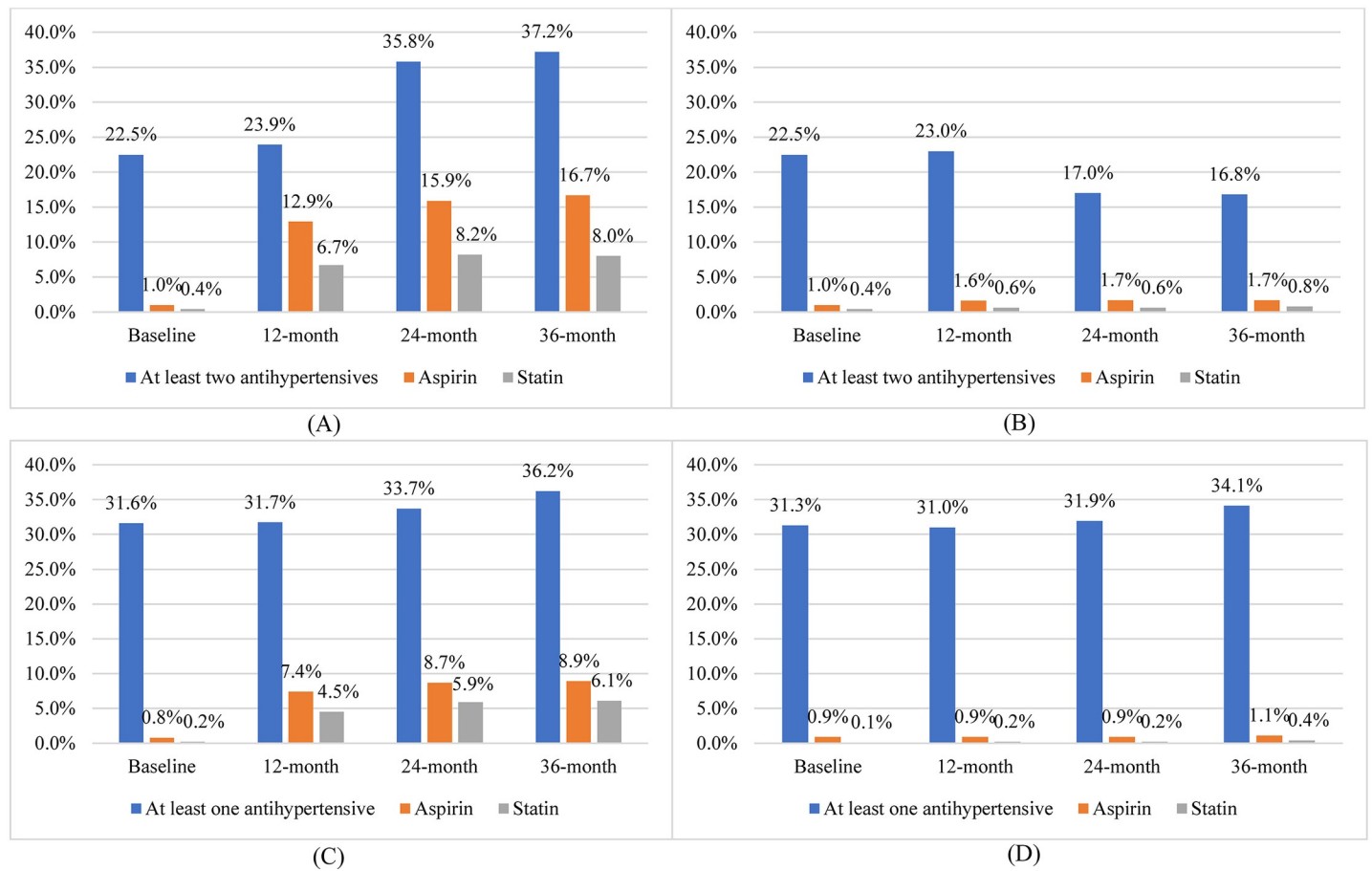

**Fig 3. Self-reported adherence to medicines.** (A and B) Self-reported adherence to medicines among patients with hypertension or hypertension and type 2 diabetes in the (A) intervention and (B) control groups. (C and D) Self-reported adherence to medicines among patients with type 2 diabetes only in the (C) intervention and (D) control groups.

36 months was not significantly higher in intervention arm patients compared to control arm patients (AOR for self-reported adherence to at least 1 antihypertensive medicine = 1.31 [95% CI: 0.93, 1.84; $P$ = 0.13]), but self-reported adherence to aspirin and/or a statin at 36 months was significantly higher in intervention arm patients compared to control arm patients (AOR for aspirin = 9.09 [95% CI: 5.44, 15.2; $P$ < 0.001]; AOR for statin = 12.8 [95% CI: 6.68, 24.5; $P$ < 0.001]). In all these subgroup analyses, 1,494 (32.6%) and 1,999 (36.2%) patients were excluded from the intervention and control arms, respectively, due to missing covariate or outcome data.

A total of 148 (1.1%) and 185 (1.3%) patients reported minor adverse events in the intervention and control arms, respectively (AOR = 0.927 [95% CI: 0.514, 1.673; $P$ = 0.802]). Of 333 reported adverse events, 134 (40.2%) were gastrointestinal discomfort, 73 (21.9%) were dizziness, 50 (15.0%) were hypodynamia, 35 (10.5%) were undefined bleeding, and 41 (12.4%) were other unclassified conditions. No significant difference was identified for any category of adverse event (Table I in S1 Text). We did not identify any serious adverse events that required emergency medical attention.

## Discussion

To our knowledge, this is the first large-scale, pragmatic, cluster randomised controlled trial with long-term follow-up that investigates the effect of a comprehensive CVD-risk-based prevention strategy on CVD events. We did not observe a statistically significant difference regarding the primary outcome measure of severe CVD events within 36 months. Nor did we identify a statistically significant effect on the primary outcome in our per-protocol analyses, where we only included intervention arm participants who reported at least 50% adherence to recommended prescriptions. However, only 4.6% of patients were retained in the intervention arm per-protocol population, despite the relatively low adherence threshold for inclusion. Therefore, along with the limitations of per-protocol analyses, to robustly understand the actual treatment effect in this context, as opposed to the effect of just providing the treatment option, would clearly require further study. There was a significant difference in the secondary outcomes of systolic and diastolic blood pressure change from baseline to follow-up that favoured the intervention over the control treatment, but it was small. Related to this, we also observed statistically significant and clinically moderate to large improvements in patient-reported adherence to 2 antihypertensives, aspirin, and statins, albeit the overall adherence rates to all medicines were low. We found no other statistically significant intervention effects for any of our other secondary outcomes within the crude or adjusted ITT or per-protocol population analyses.

Several health system issues may limit the effectiveness of such interventions. First, low health insurance coverage for medicines and high out-of-pocket payments may discourage patients from taking recommended medicines. Though our intervention substantially improved adherence rates (i.e., to 37% for recommended antihypertensives, 17% for aspirin, and 6% for statins in the intervention arm), the rates were nevertheless far lower than those in trials providing free medicines. For example, the PolyIran trial [6] achieved 81% adherence, and an atorvastatin trial [31] had 85% adherence. Based on strategies of motivating community health workers and providing free medicines, the HOPE4 trial reported 84% adherence to antihypertensives and statin [11]. Our adherence rates mimic conditions in real-life settings, where less than 50% of patients with hypertension regularly take antihypertensives [32]. In our setting, the recommended package of medicines cost US$35–US$44 per month, with participants having a median annual per capita income of US$1,424. Our process evaluation revealed that affordability of medicines was a major challenge due to low health insurance coverage for outpatient costs. In practice, most patients paid their outpatient medicines out of pocket [28,33]. China extended universal health coverage to everyone in 2009, which reduced the proportion of out-of-pocket health expenditure, but not the absolute level [34]. In addition, outpatient costs, as often seen in primary care, are proportionally much less covered than inpatient costs. This problem is exacerbated as cheaper generic medicines often become unavailable in township hospitals because the companies supplying them cannot make profits under current bidding regulations [28]. This affordability challenge has created a health equity crisis in primary care in China, wherein the poorest quintile of patients with diabetes experience the highest burden of catastrophic health expenditure [35]. Second, our participants were mainly older people who had previously been treated for hypertension or diabetes, and they were commonly reluctant to change from their existing combination tablet, which contained antihypertensive and traditional Chinese medicine, to multiple pills, which represented a higher pill burden [15]. Polypill is not available in our setting. Third, the low level of trust that patients have in family doctors in China, as we identified in our sites [28] and another province [36], may have also contributed to the low adherence to prescriptions and healthy lifestyle advice.

We designed this pragmatic trial to be embedded in real-life primary care practice so as to better inform health policy. We minimised participant eligibility criteria to reflect the general population of patients with hypertension and/or diabetes as managed under primary care. Our data collection, based on the province-wide CVD surveillance system, enabled us to collect primary and most secondary endpoints from all participants, including patients who moved out of our study sites and those who agreed to participate but soon left the trial. We were therefore able to conduct true ITT analyses without imputation, which is not possible in many trials. The pharmaceutical and healthy lifestyle interventions were designed to fit into the job descriptions of family doctors under the new payment and incentive system of China's health reform. Our studies showed that the interventions were feasible for health providers [15] and patients [17].

Our study has several limitations. First, at recruitment all participants were verbally informed that they had 20% or greater risk of developing CVD in the next 10 years, and that they needed to adopt activities to reduce this risk, such as treating their hypertension and/or diabetes and making general healthy lifestyle changes. They all had their blood pressure measured and participated in a questionnaire survey. These activities can be seen as an intervention itself as some patients in the control arm may have modified their medicines and/or lifestyles. This potentially reduced the difference in primary and secondary outcomes observed between in the intervention and control arms. Second, we employed the Asian equation to calculate the 10-year CVD risk of participants, but the Asian equation does not consider any comorbidities or complications that may contribute to a higher risk of CVD. However, we included all patients with diabetes, who were more likely to have complications than patients with hypertension. Third, there was no blinding for healthcare providers or patients, although the outcome analyses were blinded. However, given the largely null results, there was no evidence of the typical inflation of effect sizes seen due to lack of blinding. Fourth, we could not collect baseline information for the 3,196 (10%) eligible patients who declined participation based on ethical considerations. Therefore, we could not fully assess how comparable the trial cohort was to the wider target population. Fifth, our 36-month follow-up period is shorter than that of other trials employing CVD events as the primary endpoint, which normally use 60 months. However, given the health system limitations identified here, a longer follow-up period is unlikely to generate substantially better impact on CVD events. Sixth, we were not able to collect information regarding lipid profiles, blood glucose, and healthy lifestyle behaviours for all participants due to the concern that too many data collection activities could alter routine services. We collected the information in a panel of participants and will report these results elsewhere. Seventh, recall and reporting biases may exist because we collected self-reported adherence to medicines. Electronic pill boxes were not available during our study, and even if available, they may alter provider and patient behaviour. Lastly, any attempt at robustly estimating the likely effect of the treatment if adhered to was not possible due to the very small proportion of individuals who adhered at even a modest rate in the intervention arm.

Our intervention's lack of effectiveness on CVD events is consistent with results from trials examining CVD-risk-based management interventions in primary care settings in HICs [8,37–39]. One of the major limitations of these previous trials is that their interventions were all light-touch, focusing on providing healthy lifestyle advice, with only 1 suggesting antihypertensive treatment. Unlike in these trials, we observed improved adherence to medications and significant but small reductions in blood pressure. There were another 2 trials from LMICs that implemented comprehensive pharmaceutical and lifestyle interventions among populations with high CVD risk [10,11]. Both achieved reductions in blood pressure, and one in

CVD risk, but the trials were not designed to test effects on CVD events, they had a shorter duration, and their sustainability was unclear.

CVD-risk-based pharmaceutical and healthy lifestyle management remains a popular policy for primary prevention of CVD as shown in WHO recommendations [1] and certain national guidelines [40,41]. Our study, along with other studies from HICs [37–39], showed that interventions that reduce CVD risks at the population level would be unlikely to have an impact unless structural health system factors such as the availability of essential medicines are addressed first. This calls for more implementation research to examine how interventions can work in real-world settings. The context of the health system, as revealed in our study, determines the success of implementation. Health system factors at the policy, facility, and individual level need to be considered in future research. At the policy level, improved financing of primary care is required to ensure patients have adequate access to family doctor services and essential medicines. Free distribution of essential medicines, as shown in Ontario, Canada [42], improved treatment adherence, and would be expected to reduce CVD risk factors, which in return would substantially reduce societal costs associated with poor outcomes in managing diabetes and hypertension in the long run [43]. At the facility level, risk-based management guidelines need to be made compatible with local primary care delivery to accommodate regular patient follow-ups and personalised health education [44]. At the individual level, patients have to be constantly supported and encouraged for positive changes. This requires involvement of community-based health workers, patient engagement and empowerment for disease self-management, shared decision-making, and use of telemedicine and apps in the post-coronavirus-pandemic era [2].

In conclusion, our study showed that our comprehensive package of risk-based pharmaceutical and health lifestyle interventions for patients with hypertension and diabetes improved patient adherence to essential medicines and reduced patient blood pressure, but importantly it did not reduce severe CVD events. Revisiting key health system components, such as improving universal health coverage for essential medicines, is crucial in designing future trials and policies to make an impact in risk-based primary care CVD prevention.

## Supporting information

**S1 CONSORT Checklist.**
(DOCX)

**S1 Text. Supporting tables.** Table A: Comparison of primary and secondary outcome disease event rates between the intervention and control arms (cluster-level data analysis). Table B: Comparison of systolic and diastolic blood pressure outcomes between the intervention and control arms (cluster-level data analysis). Table C: Comparison of time to first CVD event and time to CVD mortality outcomes between the intervention and control arms (cluster-level data analysis). Table D: Participant characteristics of per-protocol analysis. Table E: Comparison of event rates between the intervention and control groups in the per-protocol analysis. Table F: Comparison of time to first CVD event and time to CVD mortality outcomes between the intervention and control arms in the per-protocol analysis. Table G: Comparison of systolic and diastolic blood pressure outcomes between the intervention and control arms in the per-protocol analysis. Table H: Subgroup analysis for CVD event rates. Table I: Comparison of adverse events between the intervention and control arms.
(DOCX)

## Acknowledgments

We thank the following members in the Trial Steering Committee: Prof. Kamran Siddiqi from the University of York (Chair), Prof. Yude Chen from Peking University, Prof. Stephen Leeder from the University of Sydney, and Prof. Qiang Sun from Shandong University. We thank colleagues from Zhejiang CDC and health authorities, and those from Shaoxing Prefecture, Shangyu County, Zhuji County, and Shengzhou County, and staff in the participatory township hospitals in the 3 counties. We also thank Prof. Yunxian Yu from the School of Public Health at Zhejiang University for his input in training family doctors. We also thank all research assistants and postgraduate students from Zhejiang CDC, the Chinese University of Hong Kong, and the University of Leeds who participated in the data collection and interventions.

### Transparency declaration

The lead authors affirm that the paper is an honest, accurate, and transparent account of the study being reported; that no important aspects of the study have been omitted; and that any discrepancies from the study as planned (and, if relevant, registered) have been explained.

### Patient and public involvement

Patients were involved in our process evaluation and provided valuable insights regarding how the interventions were experienced by patients. Those were published in our process evaluation paper.

### Dissemination declaration

We have involved health policy makers (MY, and others from the health bureaus) from the beginning in the activities including designing, piloting, and implementing the interventions. We also actively involved health practitioners in township hospitals and county hospitals for their comments and feedback. The results have been shared with health policy makers, doctors, and patients during the study, and have been used to improve hypertension and diabetes management in the research area and at the provincial level.

## Author Contributions

**Conceptualization:** Xiaolin Wei, Guanyang Zou, John D. Walley, Min Yu.

**Data curation:** Zhitong Zhang, Weiwei Gong.

**Formal analysis:** Zhitong Zhang, Marc K. C. Chong, Joseph P. Hicks.

**Funding acquisition:** Xiaolin Wei, John D. Walley.

**Investigation:** Zhitong Zhang.

**Methodology:** Xiaolin Wei, Marc K. C. Chong, Joseph P. Hicks, Guanyang Zou, John D. Walley.

**Project administration:** Xiaolin Wei, Zhitong Zhang, Weiwei Gong, Guanyang Zou, Jieming Zhong, Min Yu.

**Resources:** Min Yu.

**Supervision:** Xiaolin Wei, Min Yu.

**Validation:** Xiaolin Wei, Min Yu.

**Visualization:** Zhitong Zhang, Marc K. C. Chong.

**Writing – original draft:** Xiaolin Wei.

**Writing – review & editing:** Xiaolin Wei, Zhitong Zhang, Marc K. C. Chong, Joseph P. Hicks, Weiwei Gong, Guanyang Zou, Jieming Zhong, John D. Walley, Ross E. G. Upshur, Min Yu.

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
