## [Editor Report · Decision Letter 0]

22 Oct 2020

Dear Dr Wei, 

Thank you for submitting your manuscript entitled "Effect of a package of risk-based pharmaceutical and lifestyle interventions on patients with hypertension and/or diabetes in rural China: a pragmatic cluster randomised controlled trial" for consideration by PLOS Medicine.

Your manuscript has now been evaluated by the PLOS Medicine editorial staff [as well as by an academic editor with relevant expertise] and I am writing to let you know that we would like to send your submission out for external peer review.

Kind regards,

Adya Misra, PhD,

Senior Editor

PLOS Medicine

---

## [Decision Letter · Decision Letter 1]

23 Nov 2020

Dear Dr. Wei,

Thank you very much for submitting your manuscript "Effect of a package of risk-based pharmaceutical and lifestyle interventions on patients with hypertension and/or diabetes in rural China: a pragmatic cluster randomised controlled trial" (PMEDICINE-D-20-05015R1) for consideration at PLOS Medicine. 

Your paper was evaluated by three reviewers, an external academic editor, and the in-house reviewers; the reviewers' and editors' comments are enclosed below, and I hope you find them constructive. Reviewer attachments are available via this link: 

[LINK]

In light of these reviews, I am afraid that we will not be able to accept the manuscript for publication in the journal in its current form, but we would like to consider a revised version that addresses the reviewers' and editors' comments. Obviously we cannot make any decision about publication until we have seen the revised manuscript and your response, and we plan to seek re-review by one or more of the reviewers. 

We expect to receive your revised manuscript by Dec 14 2020 11:59PM. Please email us (plosmedicine@plos.org) if you have any questions or concerns.

We look forward to receiving your revised manuscript. 

Sincerely,

Emma Veitch, PhD

PLOS Medicine

On behalf of Adya Misra, PhD, Senior Editor, 

PLOS Medicine

plosmedicine.org

*Thank you for including the CONSORT flowchart and checklist with the manuscript; we also recommend you include a statement in the Methods section that the CONSORT for cluster trials guidance has been used to help in reporting the study, and include a citation/reference to the guideline (https://pubmed.ncbi.nlm.nih.gov/22951546/)

*Please update the CONSORT checklist and reupload, ensuring that you use section and paragraph numbers to indicate where the checklist item information can be found, rather than page numbers (as papers are repaginated for publication). 

Academic editor advice/comments:

I think this is quite a well done trial and the findings and explanation for the null finding are quite plausible. The failure to make a substantive increase in medication rates, and short follow-up duration probably are the main drivers of the outcome.

For the former, they have argued it is mainly due to out of pocket costs for meds which I am sure is true but their process evaluation paper suggests there might be other factors at play such as level of trust with the township clinics.

It would have been nice to see some of the financial data on out of pocket costs and whether sub-groups based on income status shed any light but they state they are planning on including that in another paper.

I agree with the conclusion that system level interventions are needed, and notably primary health care in the Community Health Centres did not play a role in this intervention.

The findings contrast with studies like HOPE-4 which did provide free medication and got quite large BP reductions (but was not powered to look at CVD events). But I think in HOPE-4 there was also stronger community health worker engagement and this may also be key along with free meds. Our group found this also in Indonesia - https://jamanetwork.com/journals/jamacardiology/fullarticle/2748764

I am not that concerned about the statin issue except that despite including it as a focus in the intervention they did not measure lipid levels but one might assume with the medication rates achieved the LDL reduction would have been small.

Comments from the reviewers:

Reviewer #1: See attachment

Michael Dewey

[see [LINK]]

Reviewer #2: 

This is a very interesting study in the important area of improving primary prevention outcomes in low-resource settings. However, the results are disappointing and it seems that lack of efficacy is mainly driven by very low adherence. Even though the adherence issue takes up most of the discussion, I think that is not enough.

First, it is not clear why adherence was assessed only by self-report, and no validated tools were used.

Second, since adherence turned out to be a crucial factor, I would suggest expanding the analysis of the reasons for non-adherence. For example, you could consider adding a logistic regression of factors predicting lack of adherence.

Aside from the adherence problem, it is not clear to me why aspirin was included in the standard drug package for patients without atherosclerotic CVD.

Finally, the text will greatly benefit from being checked by a native English speaker. It contains several mistakes in the choice of words, for example "What was this study done?"

Reviewer #3: 

The authors are reporting the results of a pragmatic trial in patients with hypertension and 10 year CVD risk of 20% or higher, and patients with diabetes regardless of their 10 year CVD risk. In this cluster design study, the intervention arm received a standardised package of medicines, individual advice on lifestyle change and adherence support. The control arm received the usual care. At 36 months, there was non-significant difference in CVD incidence rates.

1- All severe CVD events were collected through Zhejiang CDC's surveillance system: can you describe accuracy, reliability and representativeness? Are 100 % of events recorded in this system?

2- Line 258 : "in addition, family doctors asked participants if they had experienced any stroke or heart attack events that were diagnosed by hospitals at the 12th,….. This was then verified by the Validation Committee and included in the 261 records, if not previously" 

Can the authors define how many CV events were revealed by this method and not originally detected by the surveillance system.

If this was high that means other events from patients who were not seen by participating doctors for follow up may be have been missed and that would lead to a major bias in the ITT analysis.

3- Define cvd event in the paragraph starting with line 249 ,similar to how CHD and stroke were defined

4- Is it possible to describe those who declined inclusion in the trial to make sure there is no systemic bias?

5- Understand the reason for low adherence

6- In this statement: clusters and participants were successfully followed up based on electronic medical records and/or CVD events reported via family doctors: how were medical records reviewed? what percentage is electronic?

7- English and grammar need to be improved in some areas for example in the method section: line 32:

We randomised 67 township hospitals to intervention (34) or control (33) arms, and 33 recruited 31,326 participants, (15,380 in the intervention arm and 15,946 in the control 34 arm), with no known CVD, and with hypertension and a 10-year CVD risk of 20% or 35 higher, and all patients with type 2 diabetes. 

Although clear in the title, the diabetes inclusion criteria was a bit confusing here and not clear until reading table 2. An easier way to read it would be participants with diabetes, or hypertension and …/ or could say participants with hypertension and 10 -year CVD risk…. And type 2 diabetes regardless of 10-year CVD risk…

Other examples:

line 604, shows incomplete sentence:

Given the interventions were shown to be feasible for family doctors [15] and patients [17].

Line 260: incomplete sentence

This was then verified by the Validation Committee and included in the records, if not previously.

8- What were the adverse events?

9- 10 year CVD risk :

- What is the 10 year CVD risk in each arm in those with hypertension?

- Although patients with diabetes are at higher risk for CVD compared to those without diabetes, the risk of CVD is not the same in all patients. For example, the risk of CVD in diabetes may be higher in patients depending on presence of microalbuminuria, smoking, etc… so having an imbalance of those contributing factors between groups, may also lead to imbalance of CVD risk as well and therefore bias the results.

10- Line 609, would need to describe in the methods how patients were informed about their CVD risk

11- Line 635, although it makes sense, we do not have clear understanding based on this study

[LINK]

---

## [Decision Letter · Decision Letter 2]

3 Jun 2021

Dear Dr. Wei,

Thank you very much for re-submitting your manuscript "Effect of a package of risk-based pharmaceutical and lifestyle interventions on patients with hypertension and/or diabetes in rural China: a pragmatic cluster randomised controlled trial" (PMEDICINE-D-20-05015R2) for consideration at PLOS Medicine. We do apologize for the delay in our response. 

I have discussed the paper with editorial colleagues and our academic editor, and it was also seen again by three reviewers. I am pleased to tell you that, provided the remaining editorial and production issues are fully dealt with, we expect to be able to accept the paper for publication in the journal.

[LINK]

Please let me know if you have any questions, and we look forward to receiving the revised manuscript shortly.   

Sincerely,

Richard Turner, PhD

rturner@plos.org

Requests from Editors:

Please remove the word "reasonable" from the data statement (submission form). 

Please adapt the title to better match journal style: "Evaluation of a package of risk-based pharmaceutical and lifestyle interventions for patients with hypertension and/or diabetes in rural China: a pragmatic cluster randomised controlled trial".

Please add "... in [province], China ..." early in your abstract.

Please quote the study dates in your abstract.

Please quote aggregate demographic details for study participants in your abstract.

At line 39, you may wish to add a few words to define "severe CVD events".

At line 52, please adapt the sentence to: "Main study limitations include all participants being informed about their high CVD risk at baseline, non-blinding of participants, and the relatively short follow-up period available for judging potential changes in rates of CVD events." or similar.

We suggest removing the primary endpoint effect estimate from the author summary, as this is quoted in your abstract. 

At line 158 (introduction), please adapt the text to "Here we report evaluation of the relevant interventions against the trial's primary and secondary ..." or similar. 

At line 403, please correct "farmers".

Throughout the text, please adapt reference call-outs as follows: "... in HICs [3,4]." (noting the absence of spaces within the square brackets). 

Please remove the information on funding, competing interests and data sharing from the end of the main text. In the event of publication, this information will appear in the article metadata, via entries in the submission form. 

The information on ethics should be moved to the Methods section.

Please use journal name abbreviations, including "PLoS ONE" and "Lancet", consistently in your reference list.

Please rename the attached CONSORT checklist as "S1_CONSORT_Checklist" or similar, and refer to it by this name in the Methods section. 

Comments from Reviewers:

*** Reviewer #1: 

The authors have addressed my points and cleared up some of my misunderstandings.

Michael Dewey

*** Reviewer #2: 

Thank you for your thoughtful answers to my questions from the first round of review. Overall, the revised version looks significantly better. 

However, I still have couple of comments. First, I agree that your arguments for using only self-reported adherence information are quite reasonable. Nethertheless, I am convinced that this fact should be included into the list of the study limitations. 

Second, despite your considerations, aspirin is still not a standard drug for primary prevention. Given your decision to use it, I would provide information on bleedings in the intervention arm vs control.

*** Reviewer #3: 

The authors have addressed all comments appropriately and acknowledged the limitations of the study. I have no other major concerns

***

[LINK]

---

## [Editor Report · Decision Letter 3]

13 Jun 2021

Dear Dr Wei, 

On behalf of my colleagues and the Academic Editor, Dr Peiris, I am pleased to inform you that we have agreed to publish your manuscript "Evaluation of a package of risk-based pharmaceutical and lifestyle interventions for patients with hypertension and/or diabetes in rural China: a pragmatic cluster randomised controlled trial" (PMEDICINE-D-20-05015R3) in PLOS Medicine.

Prior to final acceptance, please: remove the word "reasonable" from the data statement (submission form); correct spelling of "farmers" in the abstract; and remove the information on competing interests from reference 34.

PRESS

Sincerely, 

Richard Turner, PhD 

rturner@plos.org